# Combining Therapeutic Drug Monitoring and Pharmacokinetic Modelling Deconvolutes Physiological and Environmental Sources of Variability in Clozapine Exposure

**DOI:** 10.3390/pharmaceutics14010047

**Published:** 2021-12-27

**Authors:** Kenneth H. Wills, Stephen J. Behan, Michael J. Nance, Jessica L. Dawson, Thomas M. Polasek, Ashley M. Hopkins, Madelé van Dyk, Andrew Rowland

**Affiliations:** 1College of Medicine and Public Health, Flinders University, Adelaide, SA 5042, Australia; kenny.wills@flinders.edu.au (K.H.W.); baha0004@flinders.edu.au (S.J.B.); ashley.hopkins@flinders.edu.au (A.M.H.); madele.vandyk@flinders.edu.au (M.v.D.); 2Flinders Medical Centre, Adelaide, SA 5042, Australia; michael.nance@sa.gov.au; 3SA Pharmacy, Southern Adelaide Local Health Network, Adelaide, SA 5042, Australia; jessica.dawson@sa.gov.au; 4Centre for Medicine Use and Safety, Monash University, Melbourne, VIC 3000, Australia; tom.polasek@certara.com; 5Department of Clinical Pharmacology, Royal Adelaide Hospital, Adelaide, SA 5000, Australia; 6Certara, Princeton, NJ 08540, USA

**Keywords:** therapeutic drug monitoring, clozapine, pharmacokinetic modelling, inter-individual variability, dose optimisation, medication adherence

## Abstract

**Background:** Clozapine is a key antipsychotic drug for treatment-resistant schizophrenia but exhibits highly variable pharmacokinetics and a propensity for serious adverse effects. Currently, these challenges are addressed using therapeutic drug monitoring (TDM). This study primarily sought to (i) verify the importance of covariates identified in a prior clozapine population pharmacokinetic (popPK) model in the absence of environmental covariates using physiologically based pharmacokinetic (PBPK) modelling, and then to (ii) evaluate the performance of the popPK model as an adjunct or alternative to TDM-guided dosing in an active TDM population. **Methods:** A popPK model incorporating age, metabolic activity, sex, smoking status and weight was applied to predict clozapine trough concentrations (C_min_) in a PBPK-simulated population and an active TDM population comprising 142 patients dosed to steady state at Flinders Medical Centre in Adelaide, South Australia. Post hoc analyses were performed to deconvolute the impact of physiological and environmental covariates in the TDM population. **Results:** Analysis of PBPK simulations confirmed age, cytochrome P450 1A2 activity, sex and weight as physiological covariates associated with variability in clozapine C_min_ (R^2^ = 0.7698; *p* = 0.0002). Prediction of clozapine C_min_ using a popPK model based on these covariates accounted for <5% of inter-individual variability in the TDM population. Post hoc analyses confirmed that environmental covariates accounted for a greater proportion of the variability in clozapine C_min_ in the TDM population. **Conclusions:** Variability in clozapine exposure was primarily driven by environmental covariates in an active TDM population. Pharmacokinetic modelling can be used as an adjunct to TDM to deconvolute sources of variability in clozapine exposure.

## 1. Introduction

Clozapine is the most effective antipsychotic medication for reducing both positive and negative symptoms in individuals with treatment-resistant schizophrenia (TRS) [1,2]. However, a narrow therapeutic index and high prevalence of severe toxicities, such as agranulocytosis and myocarditis, necessitate frequent intensive monitoring for the duration of treatment [3]. Despite the superior efficacy of clozapine, the drug is underutilised due to concerns regarding potential life-threatening toxicities [4,5]. Delays in clozapine initiation result in poorer patient outcomes and potentially harmful toxicities associated with antipsychotic polypharmacy [6].

Therapeutic drug monitoring (TDM) is well established as a robust approach to account for variability in drug exposure. It is performed by measuring an individual’s plasma drug concentration to assess whether this is within a recommended therapeutic concentration range, and this is then typically followed by either a dose increase, which can safely be made if the individual is below this range and without toxicity, or a dose reduction, which can confidently be made if the concentration is significantly higher than the recommended range, and toxicities are present [7]. Clozapine is a strong TDM candidate on the basis of its large (>20 fold) inter-individual variability in observed pharmacokinetics (PK), narrow therapeutic index, defined target concentration and unpredictable dose-exposure relationship [7,8]. Indeed, clozapine exposure has routinely been assessed by TDM for many years [9], and a robust therapeutic concentration range of 350 to 800 ng/mL has been established [10]. The application of TDM-guided clozapine dosing has been demonstrated to reduce the incidence of drug related toxicities [11], improve efficacy and reduce the incidence of relapse [12]. Although clozapine’s primary metabolite, norclozapine, has no therapeutic activity, measurement of the ratio of clozapine to norclozapine has potential clinical utility. Specifically, a clozapine to norclozapine ratio < 0.67 can identify patients with a CYP1A2 rapid metaboliser phenotype, poor adherence in the 24 h prior to assessment, or those who are heavy smokers, while a clozapine to norclozapine ratio > 1.5 can identify patients with a CYP1A2 poor metaboliser phenotype, inhibitory drug interactions, chronic poor adherence, or a non-trough TDM sample [13,14]. While generally supportive, a recent review highlighted limitations to monitoring the clozapine to norclozapine ratio in isolation [15], and as such the exact role of the ratio in clinical practice remains to be determined. Specifically, regarding the relationship between adherence and the clozapine to norclozapine ratio, as norclozapine is formed from clozapine and has a longer half-life, poor adherence immediately prior to assessment results in a low clozapine concentration compared to norclozapine and a reduced clozapine to norclozapine ratio. In contrast, chronic poor adherence results in a relatively low norclozapine concentration in patients who are adherent immediately prior to assessment, and this results in an elevated clozapine to norclozapine ratio, as there is insufficient time for norclozapine to return to steady state.

Model-informed precision dosing (MIPD) is an approach for optimising drug dosing [16]. Typically based on a population pharmacokinetic (popPK) model, it offers the potential of improved initial clozapine dose selection and dose optimisation, thereby minimising the duration of sub- or supra-therapeutic clozapine concentrations. MIPD has demonstrated potential applications in multiple therapeutic domains, including oncology [17,18,19]. Variability in clozapine exposure has been associated with differences in the activity of cytochrome P450 (CYP) 1A2 (the major enzyme responsible for the clearance of clozapine), age, sex, weight, height, body mass index (BMI), ethnicity, smoking status and the use of concomitant medication [20,21]. A popPK model built using a multiple linear regression model that accounted for subject age, sex, metabolic activity and smoking status defined up to 82% of the inter-individual variability in clozapine trough concentration (C_min_) in a cohort of 3782 patients [22]. However, this popPK model has not been applied to support MIPD for clozapine [8]. Instead, initiation with a conservative initial dose and careful up-titration with rigorous TDM remains the accepted standard (weekly for the first six months, fortnightly for the next six months and monthly from one year on).

Physiologically based pharmacokinetic (PBPK) modelling and simulation allows for the simulation of drug pharmacokinetics based on the physiological covariates of a population, the physiochemical and in vitro metabolic characteristics of a drug and the dosing conditions. PBPK modelling is routinely utilised in drug development to support the transition of a program from pre-clinical to early phase clinical studies. In recent years, the capacity of PBPK modelling to evaluate physiological covariates associated with variability in drug exposure has gained attention [17,18,23,24,25]. Specifically, regarding the dosing of anti-psychotic medicines, Polasek et al. (2018) demonstrated that an individual’s steady state olanzapine concentration could be predicted using a PBPK model that accounted for covariates that influence olanzapine pharmacokinetics. Thus, PBPK has the potential to be applied as a MIPD approach in clinical practice. 

This study employed three interrelated but distinct platforms that account for pharmacokinetic variability (popPK modelling, PBPK modelling and TDM) to deconvolute sources of variability in clozapine exposure and define an optimal strategy to guide clozapine dosing. The specific objectives of the study were to (i) verify the importance of dose and physiological covariates identified in the popPK model reported by Rostami et al. (2004) in a population free from environmental covariates using PBPK modelling, (ii) define the relative importance of physiological versus environmental covariates as sources of inter-individual variability in clozapine exposure, and (iii) define the optimal role of the popPK model as an adjunct or alternative to TDM-guided dosing in an active clozapine TDM population. 

## 2. Materials and Methods

### 2.1. Physiologically Based Modelling and Simulation

PBPK simulations were performed using the Simcyp population-based simulator (version 19.1; Certara, Sheffield, UK) [26]. The differential equations used by the simulator describing enzyme kinetics and the impact of covariates have been described previously [27].

PBPK simulations used the in-built clozapine compound file (Sim-Clozapine) [26]. Clozapine area under the plasma concentration time-curve (AUC) and C_min_ were simulated using a ‘minimal PBPK model’ comprising a liver compartment and a merged compartment representing all other organs [28,29,30]. PBPK simulations undertaken to evaluate the importance of physiological covariates reported in the popPK model were performed daily at doses between 200 and 600 mg. As there is no specific input field for smoking status as a covariate in Simcyp, simulations assessed CYP1A2 abundance as a combined metric to account for basal metabolic activity (clozapine to norclozapine ratio) and smoking status. 

The importance of dose as a covariate influencing clozapine exposure was evaluated in PBPK simulations (free from environmental covariates) and in the observed clinical data from the TDM population. In order to directly compare the importance of dose between the PBPK simulations and TDM population subjects, PBPK simulations were matched to the TDM population for age, gender, and clozapine dose as follows: cohort 1 (*n* = 9; 31–63 years, 44% female, 200 mg), cohort 2 (*n* = 26; 21–59 years, 27% female, 300 mg), cohort 3 (*n* = 20, 27–60 years, 10% female, 400 mg), cohort 4 (*n* = 16, 28–63 years, 56% female, 500 mg) and cohort 5 (*n* = 7, 28–63 years, 0% female, 600 mg). Simulations were performed with oral dosing daily at 9:00 am for 7 days, with ten virtual trials performed in each cohort. The full study workflow is described in Figure 1. 

### 2.2. Observed Clinical Data

The performance of the popPK model was assessed in an active clozapine TDM population comprising 142 subjects (27% female) dosed to steady state (>7 days) at Flinders Medical Centre, Adelaide, South Australia (Table 1). Data were collected for patients treated with clozapine during a 12-month period from November 2019 to October 2020. Patient demographics (sex, age, weight, height, BMI and smoking status) and covariates describing steady state clozapine exposure (dose, clozapine C_min_ and norclozapine C_min_) were obtained through electronic health records. Clozapine to norclozapine ratio was calculated as clozapine C_min_ divided by norclozapine C_min_. Patients had an average of 15 (range 6 to 39) clozapine TDM results during the analysis period; to avoid bias within the dataset, only the most recent TDM result for each patient, obtained following ≥7 days stable dosing, was included in the analysis. Blood (K_2_EDTA) samples for analysis of clozapine trough concentration were collected 20 to 24 h following the most recent clozapine dose. Clozapine and norclozapine concentrations were quantified using a validated liquid chromatography–mass spectrometry assay approved for clinical TDM testing and reported by the South Australian state services for routine diagnostic and clinical pathology testing (SA Pathology Special Chemistry Directorate), which is accredited by the National Association of Testing Authorities (NATA) of Australia. Access to participant health records was approved by the Southern Adelaide Clinical Human Research Ethics Committee (SACHREC; approval id 200.17, approved October 2017).

### 2.3. Population Pharmacokinetic Model

This study employed a published clozapine popPK model that was built by Rostami et al. (2004) by stepwise backward multiple regression analysis. Verification data associated with this model are included in the original publication. The equation underpinning this popPK model is:*Log10 (C) = 0.811 log10 (dose) + 0.332 (MR) + 0.06941 (sex) + 0.002263 (age)*
*+ 0.001976 (weight) − 0.171 (smoking) − 3.180*
where dose is mg/day; sex is male = 0, female = 1; smoking is non-smoker = 0, smoker = 1; weight is kg; MR is plasma clozapine to norclozapine ratio.

### 2.4. Statistical Analysis

Data from PBPK-simulated and observed TDM populations are presented as the geometric mean and 95% confidence interval (CI). As it was not possible to assume equivalent standard deviations (SD) between cohorts, Brown–Forsythe ANOVAs were performed to assess the statistical significance of the impact of dose on clozapine exposure, with Dunnett’s T3 multiple comparison testing performed to assess differences in exposure between individual dose levels. One-way ANOVA with Dunnett’s T3 multiple comparison testing was similarly used to evaluate differences in the physiological factors reported to influence clozapine exposure in the full TDM population and each of the associated dose cohorts. 

Analyses were performed to quantify the performance of the popPK model with respect to predicting the log transformed clozapine C_min_ for individual subjects in PBPK simulations and a TDM population. The popPK model incorporated clozapine dose, age, sex, weight, CYP1A2 function (as CYP1A2 abundance in PBPK simulations and smoking status combine with clozapine to norclozapine ratio in the TDM population). Probability (P) values lower than 0.05 were considered statistically significant. PopPK model performance was defined based on the coefficient of determination (r^2^) and significance (P).

## 3. Results

### 3.1. TDM Population Demographics

Demographic data relevant to clozapine exposure are reported in Table 1 for the full TDM population (*n* = 142) and each of the individual dose cohorts. There were no significant differences in covariates across the dose cohorts (ANOVA *p* ≥ 0.29) or between the full cohort and any individual dose cohort (Dunnett’s multiple comparison test *p* ≥ 0.36). One hundred and thirty-nine patients (97.8%) had a clozapine concentration within the accepted target concentration range (350 to 800 ng/mL). Compared to the population investigated by Rostami et al. 2004, the current TDM population was older: 42 (21 to 69) years versus 36 (24 to 50) years, and heavier: 95 (37 to 176) kg versus 80 (60 to 102) kg. The mean (range) clozapine dose of 366 (100 to 800) mg/day and C_min_ 468 (192 to 950) ng/mL in the current study were consistent with the prior study: 460 (250 to 700) mg/day, and 430 (110 to 860) ng/mL, respectively. The clozapine to norclozapine metabolic ratio in the current study, 2.22 (0.84 to 4.38), was significantly higher compared to the prior study’s ratio of 1.32 (0.69 to 2.07).

### 3.2. Investigation of Dose as a Determinant of Clozapine Exposure

The relationship between clozapine dose and clozapine C_min_ was evaluated in the TDM and PBPK-simulated populations. The relationship between clozapine dose and the clozapine to norclozapine ratio was also evaluated in the TDM population. Mean (±95% CI) data describing clozapine C_min_ and the clozapine to norclozapine ratio in the TDM population are presented in Table 2. A mean increase in clozapine C_min_ of 52.5 ± 16.9 ng/mL per 100 mg dose increment was observed in the PBPK-simulated population, which is consistent with the reported dose proportional pharmacokinetics of clozapine [31]. No trend in clozapine C_min_ across dose increments was observed in the TDM population (Table 2).

Analysis of PBPK simulations performed at daily clozapine doses ranging from 200 to 600 mg demonstrated a significant difference in clozapine concentration across these doses (*p* < 0.001). Multiple comparison testing demonstrated that clozapine exposure differed significantly between doses of 200 mg and 500 to 600 mg (*p* <0.001) and between doses of 300 mg and 500 to 600 mg (*p* < 0.001). Differences in clozapine concentration between the 200 and 300 mg dose, the 500 and 600 mg dose, and the 400 mg dose with any other dose were insignificant (*p* > 0.25). In contrast to the dose-related changes in clozapine concentration observed between the PBPK-simulated cohorts, analysis of the TDM population demonstrated that while a statistically significant difference in exposure (ANOVA *p* = 0.03) was observed across the 200 to 600 mg dose range (Figure 2), differences between individual dose levels were non-significant (*p* > 0.16). There was no association between clozapine dose and the clozapine to norclozapine ratio in the TDM population (*p* = 0.095; R^2^ = 0.020) (Figure 3). Similarly, no difference (*p* = 0.54) in the clozapine to norclozapine ratio was observed between the dose cohorts within the TDM population.

### 3.3. Investigation of Physiological Covariates Influencing Clozapine Exposure

Consistent with the reported associations of age, metabolic activity, sex and weight with clozapine exposure, in the PBPK-simulated population, multiple linear regression modelling demonstrated that sex, age, weight and CYP1A2 abundance predicted the log transformed clozapine C_min_ with an R^2^ of 0.7698. These data support the physiological basis of the popPK model proposed by Rostami et al. 2004, and indicate that under optimal conditions, and by accounting for these covariates, it should be possible to account for approximately 77% of inter-individual variability in clozapine exposure (Figure 4). Notably, univariable analyses in the PBPK-simulated population demonstrated that sex (*p* = 0.0002) and CYP1A2 abundance (*p* < 0.001; Figure 5A), but not age or weight (*p* ≥ 0.168) were independently significantly associated with clozapine C_min_.

### 3.4. Application of the popPK Model to a TDM Population 

In contrast to the strong correlation observed in the PBPK-simulated population, in the TDM population, the predicted clozapine C_min_ based on the popPK model did not correlate with the observed C_min_. The correlation between popPK-predicted and observed C_min_ was equivalently poor across the full (*n* = 142; R^2^ = 0.049) and stratified dose (*n* = 78; R^2^ = 0.042) populations. The popPK-model-predicted clozapine C_min_ was >1.5-fold higher than the observed C_min_ in 69% of patients (Figure 6) and exceeded the 800 ng/mL upper threshold of the target concentration range in 52% of patients. As shown in Figure 7, in the TDM population, the difference between popPK-predicted and observed clozapine C_min_ was strongly correlated (*p* < 0.0001, R^2^ = 0.597) with the clozapine to norclozapine ratio, which may be a marker of poor adherence or inhibitory drug interaction. Given the high proportion of patients with a clozapine to norclozapine ratio > 1.5 (93%), it is unlikely that this is reflective of basal CYP1A2 poor metaboliser status. The difference in popPK-predicted versus observed clozapine C_min_ was not associated with any of the other physiological covariates included in the popPK model. R^2^ values for the association of other physiological covariates with the difference in predicted and observed clozapine C_min_ were 0.089, 0.0008, 0.0032, 0.0123 and 0.0025 for dose, sex, age, weight and smoking status, respectively. 

Consistent with the lack of correlation between the observed C_min_ and the popPK-predicted C_min_, each of the individual covariates included in the popPK model (sex, age, weight, clozapine to norclozapine ratio and dose) similarly demonstrated a lack of association with the observed clozapine C_min_ (*p* > 0.2). Of particular interest, the R^2^ for the clozapine to norclozapine ratio (a phenotype trait for CYP1A2) was 0.008 (Figure 5B); this is in contrast to the strong performance of CYP1A2 abundance (R^2^ = 0.7698) in the PBPK-simulated population. 

### 3.5. Post Hoc Analyses

Post-hoc subgroup analyses in non-obese (*n* = 64; R^2^ = 0.097) and age < 50 years (*n* = 108; R^2^ = 0.075) individuals demonstrated modest independent improvements in the predictive performance of the popPK model, with the strongest correlation observed in non-obese individuals aged < 50 years (R^2^ = 0.172). In contrast, post-hoc subgroup analysis of individuals with a clozapine to norclozapine ratio < 1.5 (*n* = 19) demonstrated a marked improvement in the performance of the popPK model. In this subgroup, the performance of the popPK model with respect to predicting clozapine C_min_ (R^2^ = 0.489, *p* = 0.0009) was comparable to the previously reported performance for this model (Figure 8).

## 4. Discussion

This study demonstrates that in an active TDM population, physiological differences account for a small portion of observed variability in clozapine exposure, and the primary function of TDM is to account for environmental covariates. Specifically, by applying the popPK model of Rostami et al. (2004) to the output of PBPK simulations, it was confirmed that, in the absence of environmental covariates, accounting for physiological covariates defined >75% of inter-individual variability in clozapine exposure. This PBPK simulation analysis defined the optimal possible performance of the popPK model with respect to describing inter-individual variability in clozapine exposure. The impact of environmental covariates was then assessed by comparing the predicted clozapine exposure based on the popPK model to the observed exposure in an active TDM population. 

Understanding the contribution of physiological versus environmental covariates as drivers of variability in clozapine PK defines the capacity of precision dosing and the optimal approach to employ to guide dosing. Specifically, when variability is predominantly driven by physiological covariates (such as age, sex and weight), an individual’s exposure is predictable based on a model that accounts for these covariates, and is likely to remain more stable over time. In this setting, prospective dose selection using MIPD with sporadic on treatment TDM is the optimal approach for precision dosing, as it will minimise the initial time taken to achieve optimal exposure and minimise the ongoing logistical burden associated with monitoring. In contrast, when variability is predominantly driven by environmental covariates (such as adherence, diet, and complex drug interactions), an individual’s exposure is less predictable, and is likely to fluctuate more over time. In this setting, prospective dose selection is unlikely to provide meaningful benefit, and intensive on treatment TDM supplemented with MIPD to deconvolute the source of variability is the optimal approach. Data presented in Figure 4, Figure 5 and Figure 6 support the reported impact of covariates, including age, sex, weight, smoking status and CYP1A2 activity, on clozapine exposure, but demonstrate that in an active TDM population, environmental factors such as adherence or inhibitory drug interactions play a significant role in determining clozapine exposure. 

When clozapine dose and physiological covariates affecting clozapine exposure were accounted for, the popPK model predicted that the clozapine C_min_ was >1.5-fold higher than the observed C_min_ in 69% of patients (Figure 6). The overprediction of clozapine C_min_ by the popPK model indicates that in this TDM population, either a high proportion of patients cleared clozapine at an increased rate, were poorly adherent or experienced inhibitory drug interactions. Post hoc analysis combining the popPK-model-predicted exposure with interpretation of the clozapine to norclozapine ratio deconvoluted these two possibilities. Increased clozapine clearance are associated with a low clozapine to norclozapine ratio (<0.67), an association which should be more pronounced at higher clozapine doses, while poor adherence or inhibitory drug interactions were associated with a high clozapine to norclozapine ratio (>1.5), with this association being independent of dose. The mean (range) clozapine to norclozapine ratio in the current study of 2.22 (0.84 to 4.38) was significantly higher than the threshold indicating poor medication adherence (>1.5), and was not associated with dose requirement (R^2^ = 0.0198; Figure 3a. Notably, in the TDM population, the difference between the popPK-model-predicted and observed clozapine C_min_ was strongly positively correlated (*p* < 0.0001, R^2^ = 0.597) with the clozapine to norclozapine ratio but no other physiological covariate included in the popPK model (Figure 7). A post hoc subgroup analysis in individuals with a clozapine to norclozapine ratio < 1.5 further demonstrated a marked improvement in performance for the popPK model (R^2^ = 0.489; Figure 8), comparable to the reported performance for this model. These data represent the strongest possible evidence of widespread poor medication adherence within this clozapine TDM population, despite the apparent robust proportion of patients with a clozapine C_min_ within the target therapeutic range. 

While clozapine and norclozapine concentrations were routinely quantified and reported by the hospital TDM service, medication adherence was clinically assessed primarily on the basis of clozapine C_min_. Indeed, it is possible that for many patients, poor medication adherence in a setting of a clinician targeting a clozapine C_min_ within the range of 350 to 800 ng/mL results in a higher than necessary clozapine dose, with the significance of this being that if the patient is placed in a setting where the poor medication adherence (or other environmental factor reducing clozapine exposure) is resolved, the patient will be placed at greater risk of clozapine toxicity. Indeed, the popPK model demonstrated that in the absence of environmental covariates, the predicted clozapine C_min_ based on the patient’s current dose would exceed the 800 ng/mL upper threshold of the target concentration range in 52% of patients.

It is important to acknowledge that while the TDM population studied here (Table 1) was comparable to the population used to develop the original clozapine popPK model [22] in many aspects, patients were, on average, 15 kg heavier and 6 years older. Indeed, 84% of the current TDM population were overweight (BMI > 25 kg/m^2^), while 54% were obese (BMI > 30 kg/m^2^). It has recently been demonstrated that all aspects of clozapine pharmacokinetics (absorption, distribution, metabolism and excretion) are perturbed in overweight and obese individuals [32], a phenomenon that is observed for many drugs [33,34]. Given the high propensity for clozapine to induce metabolic disturbances that result in profound weight gain [35,36], consideration of the potential impact of on-treatment weight gain on long term clozapine exposure warrants consideration. Similarly, clozapine pharmacokinetics are known to be altered with increasing age [37,38]. While it is conceivable that the greater prevalence of obese and older patients in the current TDM population contributed to the lack of correlation between predicted and observed clozapine C_min_ in the full population, sensitivity analyses demonstrated only modest improvements in correlation in the sub-group (*n* = 58) of non-obese individuals < 50 years (R^2^ 0.049 to 0.172). In this sub-group, the popPK model still only accounted for 17% of the variability in clozapine C_min_, which remained considerably lower than the optimally achievable 77% of variability in the PBPK simulation population, the 49% of variability accounted for in the sub-group of individuals with a clozapine to norclozapine ratio < 1.5 in the current population, and the 48% of variability accounted for in the previously studied population. 

Although TDM can detect treatment failure and arising toxicity at an early and potentially preventative stage by accounting for physiological and pharmacological factors, this study highlights that the benefit of TDM extends beyond this. TDM has the ability to identify valuable information regarding harmful drug–drug interactions and treatment adherence, which can trigger clinician and patient education, respectively, leading to the safer and more effective use of drugs. Additionally, unintentional or intentional environmental or lifestyle factors such as diet/food intake or uncontrolled supplement/herbal product intake can also be detected. It is worth noting that substance abuse is frequently reported in this patient population [39]. Notably, these factors can be readily detected using alternate TDM platforms/assays. In recent years, biomarker and predictive modelling strategies have been presented as an attractive alternative to TDM-guided dosing [18,23,24,40,41], with the pretence underpinning these approaches being that by prospectively accounting for physiological covariates associated with variability in exposure, it may be possible to predict an individual’s dose requirement. This study demonstrates the potential limitations of such approaches if environmental covariates such as poor medication adherence play a significant role in determining drug exposure and if these are therefore not considered. In this setting, MIPD may still play a role as an adjunct to TDM-guided dosing by assisting in deconvoluting the sources of variability in exposure and guiding appropriate interventions such as clinician and patient education. This finding has potentially significant implications for the application of MIPD in other settings such as oncology, where medication adherence may be low due to medication-related toxicities, and emphasises the importance of TDM as a strategy to monitor exposure in these settings [7,42]. 

## 5. Conclusions

In conclusion, data presented here demonstrate the continued importance of TDM for clozapine as the gold standard for individualising and monitoring clozapine dosing. MIPD may serve as a useful adjunct to TDM in patients who do not respond as expected to a given clozapine dose, specifically to assist in deconvoluting the primary driver of sub-optimal exposure. 

## Figures and Tables

**Figure 1 pharmaceutics-14-00047-f001:**
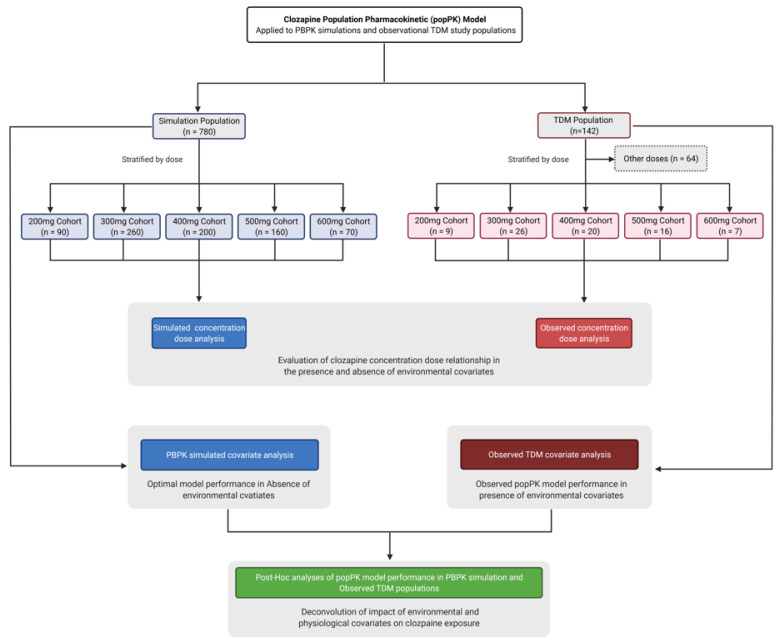
Study design and workflow for simulated and TDM populations.

**Figure 2 pharmaceutics-14-00047-f002:**
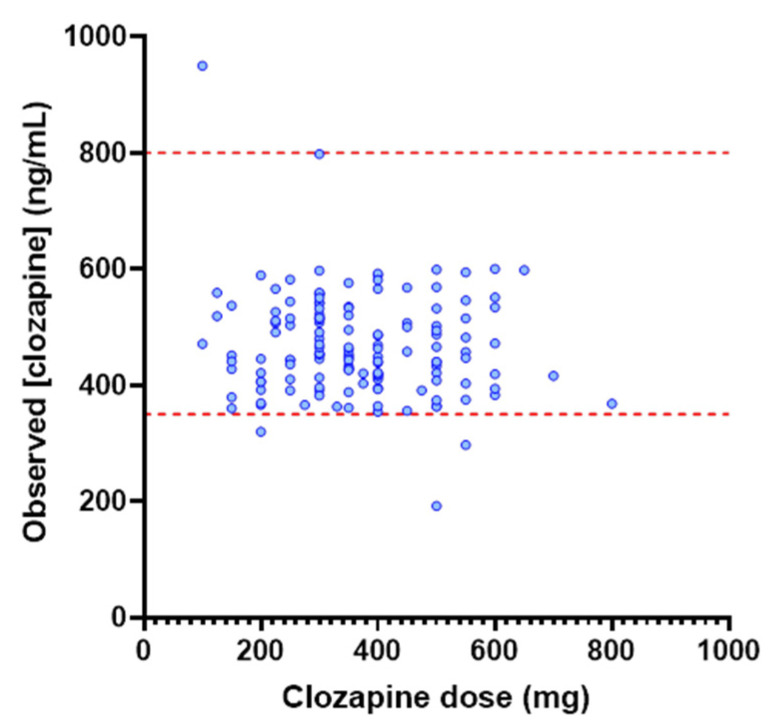
Relationship between clozapine dose and C_min_ in the TDM population (*n* = 142). Red dash lines indicate lower and upper limits of target concentration range (350 to 800 ng/mL).

**Figure 3 pharmaceutics-14-00047-f003:**
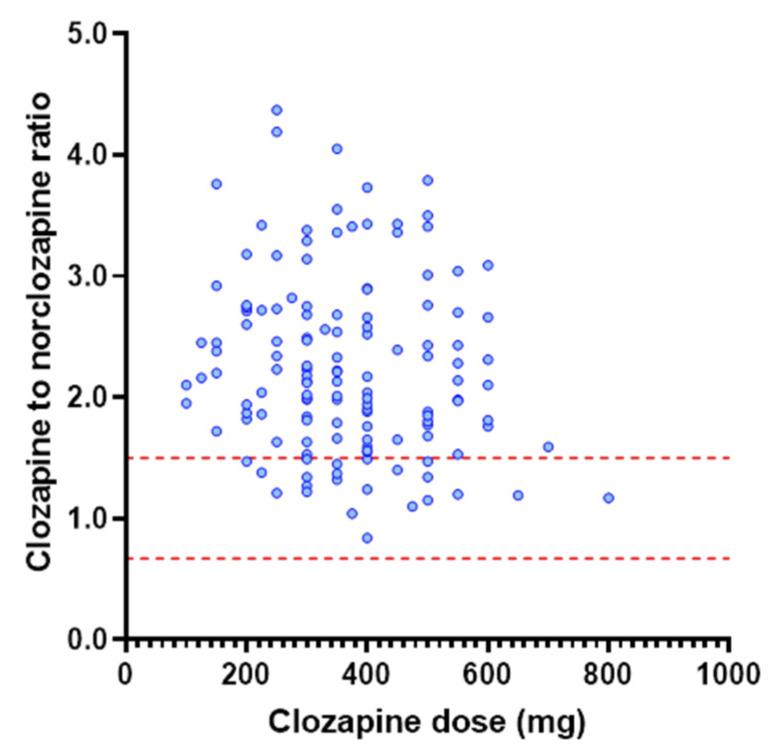
Relationship between clozapine dose requirement and clozapine to norclozapine ratio in the TDM population (*n* = 142). Red dash lines indicate lower and upper range of clozapine to norclozapine ratio (0.67 to 1.5) associated with normal CYP1A2 activity and robust adherence.

**Figure 4 pharmaceutics-14-00047-f004:**
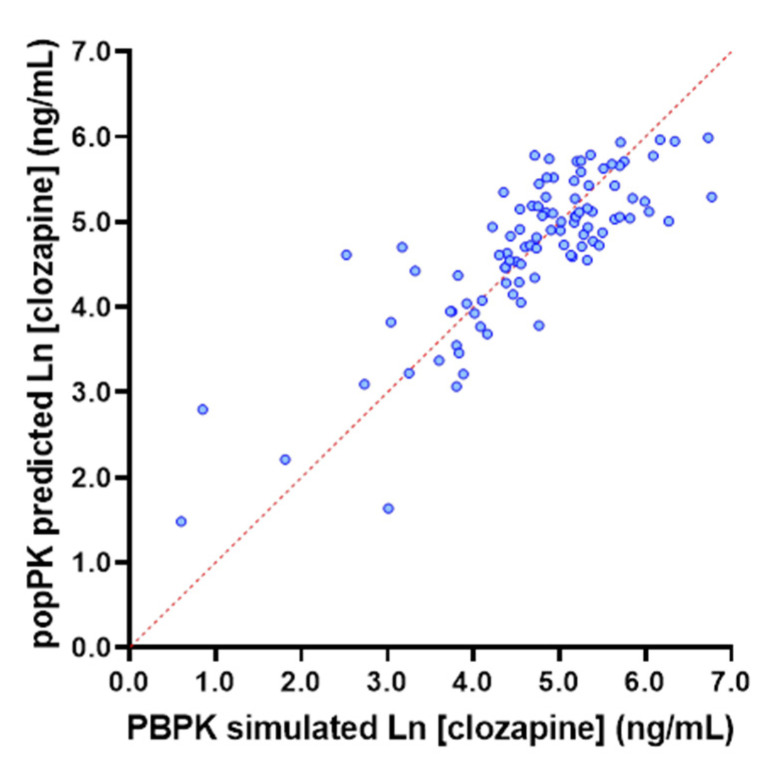
Performance of popPK model based on age, CYP1A2 abundance, sex and weight with respect to describing log transformed clozapine C_min_ in the PBPK-simulated population (*n* = 780). Red dash line indicates line of identity.

**Figure 5 pharmaceutics-14-00047-f005:**
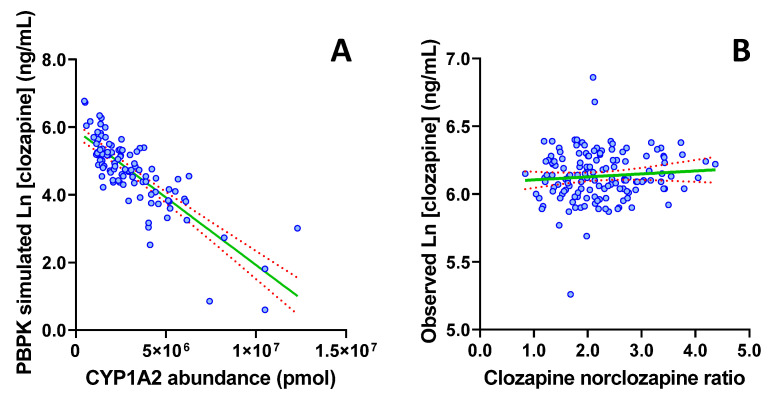
Relationship between markers of CYP1A2 function and log transformed clozapine trough concentration. Panel (**A**); CYP1A2 abundance in PBPK-simulated population (*n* = 780), Panel (**B**); clozapine to norclozapine ratio in TDM population (*n* = 142).

**Figure 6 pharmaceutics-14-00047-f006:**
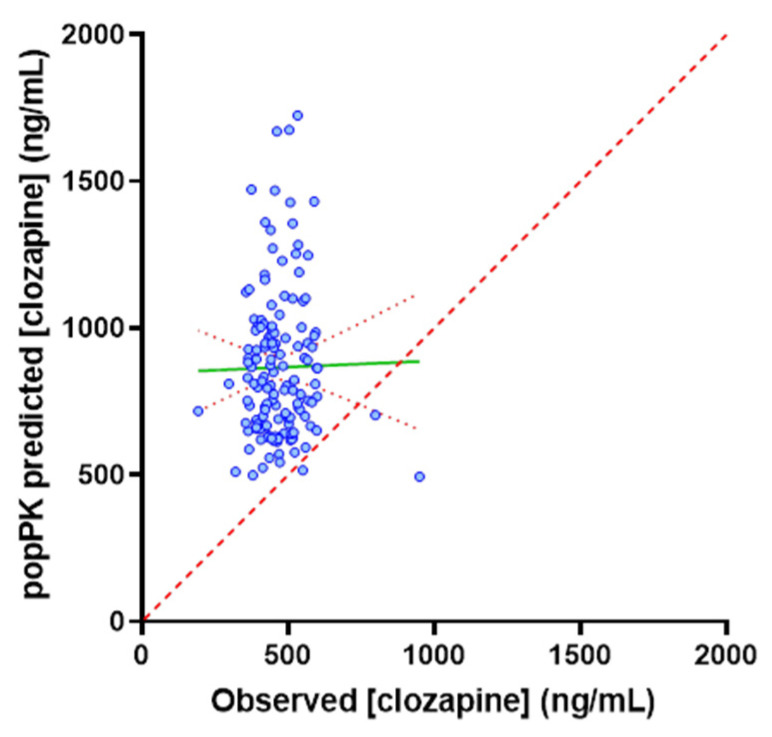
Correlation between popPK-predicted and observed clozapine C_min_ in the TDM population (*n* = 142). Red dash line indicates line of identity.

**Figure 7 pharmaceutics-14-00047-f007:**
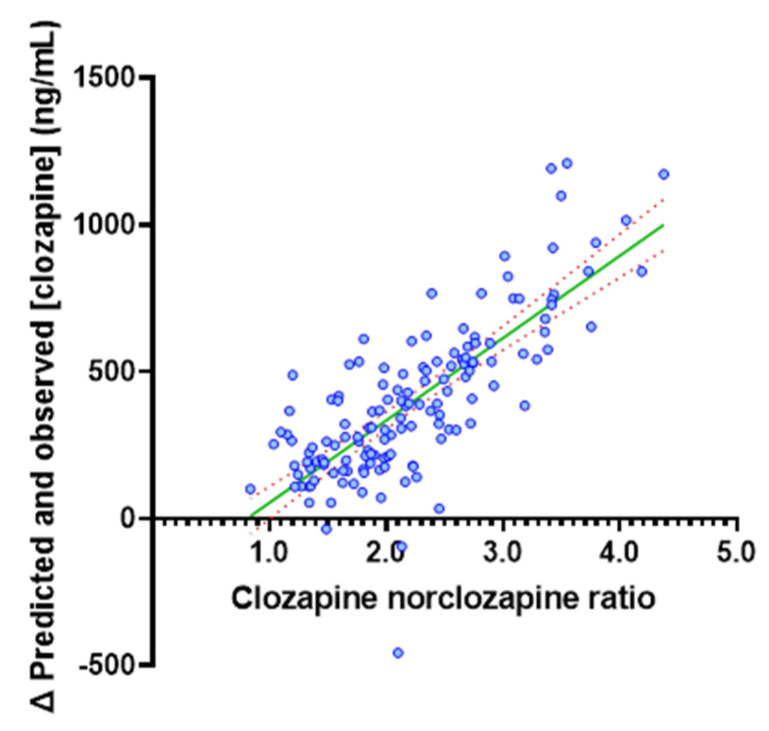
Association between the difference (Δ) in popPK-predicted to observed clozapine C_min_ and clozapine to norclozapine ratio in the TDM population (*n* = 142).

**Figure 8 pharmaceutics-14-00047-f008:**
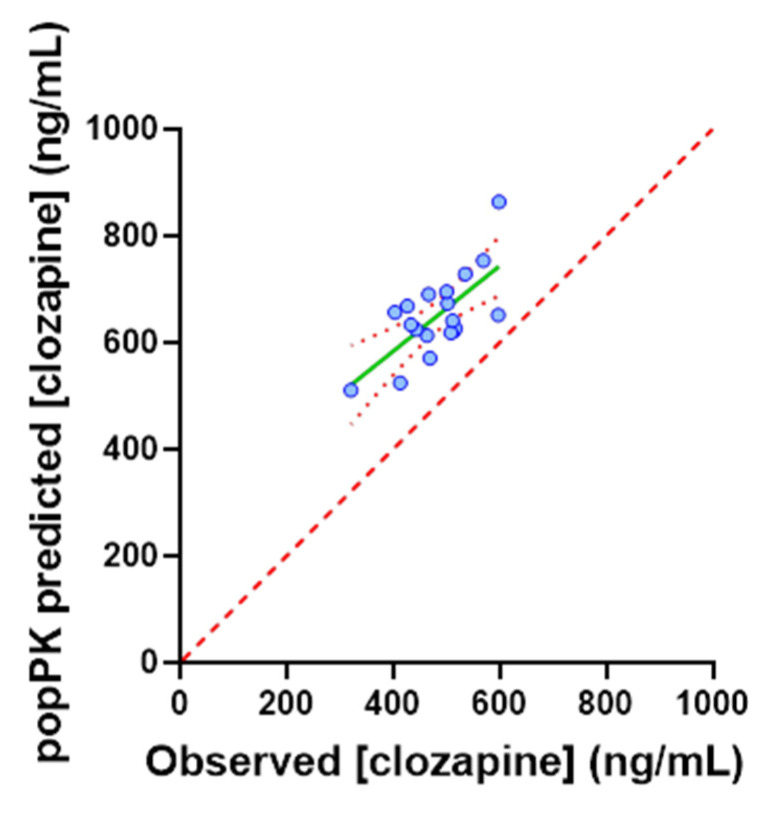
Post hoc analysis correlating popPK-predicted with observed clozapine Cmin in the subgroup of the TDM population with a clozapine to norclozapine ratio < 1.5 (*n* = 19). Red dash line indicates line of identity.

**Table 1 pharmaceutics-14-00047-t001:** Demographic parameters describing the full TDM population and dose level cohorts. Data presented as mean (range).

	Full	200 mg	300 mg	400 mg	500 mg	600 mg	Sig.
Subjects (n)	142	9	26	20	16	7	
Dose (mg)	366 (100–800)	200	300	400	500	600	
Sex (% female)	27	44	27	10	56	0	
Age (years)	42 (21–69)	(46(21–63)	38 (21–59)	41 (27–60)	38 (28–63)	40 (28–63)	0.29
Weight (kg)	95 (37–176)	94 (65–109)	88 (59–162)	99 (60–146)	91 (68–120)	102 (82–119)	0.46
BMI (kg/m^2^)	31(18–48)	32 (23–37)	29 (18–48)	30 (18–42)	31 (19–45)	32 (25–37)	0.65
Smoker (%)	65	44	62	90	75	86	

**Table 2 pharmaceutics-14-00047-t002:** Parameters describing clozapine exposure in each dose level cohort from the TDM population.

Population	200 mg		300 mg		400 mg		500 mg		600 mg	
	Mean	95% CI	Mean	95% CI	Mean	95% CI	Mean	95% CI	Mean	95% CI
Clozapine tough concentration (ng/mL)	413	355–471	504	470–537	449	418–481	443	392–494	479	401–557
Clozapine to norclozapine ratio	2.34	1.90–2.79	2.08	1.83–2.33	2.12	1.77–2.46	2.22	1.77–2.67	2.34	1.89–2.79
% Therapeutic	89		100		100		94		100	

## Data Availability

Deidentified individual participant data used to support this study will be made available following submission of a valid research plan that is within scope to the corresponding author.

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
