# Peer review of "Combining Therapeutic Drug Monitoring and Pharmacokinetic Modelling Deconvolutes Physiological and Environmental Sources of Variability in Clozapine Exposure"

_pharmaceutics, 2021, doi:10.3390/pharmaceutics14010047_

Round 1
Reviewer 1 Report
The authors of the manuscript present three strategies to perform model-informed precision dosing of clozapine (PBPK, TDM and PopPK). The main goals were to assess the relative importance of physiological and environmental covariates as potential sources of inter-individual variability of clozapine exposure, as well as the complementary or alternative use of PopPK to TDM. The work is original, detailed and well-written. The authors obtained a good correlation between clozapine concentrations predicted by PBPK simulation and the PopPK model of Rostami-Hodjegan et al., both based on physiological variables (e.g., age, sex, weight). Nevertheless, there were differences between observed Cmin (TDM) and predicted Cmin (PopPK). As such, TDM accompanied by MIPD was recommended when variability is caused by environmental covariates. I have only a few questions:
1. In lines 60-65, the authors mention that a clozapine/norclozapine ratio > 1.5 suggests poor medication adherence based on reference 13 (Couchman et al. 2010). This is also said in the discussion “poor medication adherence is associated with a high clozapine to norclozapine ratio (> 1.5)” (lines 321-322). However, Couchman et al. associated poor adherence with a clozapine/norclozapine ratio < 0.5. Please clarify this aspect and provide a proper bibliographic reference. Furthermore, include a brief explanation to readers of how clozapine/norclozapine ratios relate to treatment adherence.
2. Line 251-254. The authors refer the relationship between clozapine/norclozapine ratios and adherence once more, and use it to justify the difference between observed Cmin (TDM) and predicted Cmin (PopPK). Is it because adherence is considered an environmental variable that is not included in the PopPK model? Are there other environmental variables that could account for the discrepancy? If so, why did the authors consider that adherence is the predominant one? This should be further explored in the manuscript.
3. Later, in lines 276-278, the relationship between PopPK and TDM improves in a subgroup analysis of individuals with a clozapine to norclozapine ratio < 1.5. According to the authors, would this indicate that PopPK predictions are better correlated with TDM observations in patient groups with higher treatment adherence? Moreover, would it imply that only 19 out of 142 individuals (TDM population) adhere to therapy? Does this not significantly contrast with the high percentage of patients with clozapine concentrations within the target therapeutic range (139 patients: 97.8%)?
Minor aspect:
Line 78 – First abbreviation of trough concentration (Cmin). Repeated again in line 112.
Author Response
We thank the Reviewer for their time, please see the attached file.

Reviewer 2 Report
In this manuscript, Wills and colleagues present a comparison of the performance of a popPK model for clozapine Cmin concentrations developed using physiologically based pharmacokinetic (PBPK) modelling using data from a 2004 analysis in the UK and Eire by Rostami-Hodjegan to concentrations obtained for a current population in Australia enrolled in active therapeutic drug monitoring (TDM) for clozapine. While simulations of clozapine Cmin using the model confirmed that the model adequately accounted for variability in clozapine concentrations based on the included physiological covariates, it did not well-predict the variability observed in the measured clozapine concentrations observed in the TDM population. Overall the model overpredicted clozapine Cmin values. The authors speculated that this variability could be accounted for by environmental covariates, namely medication compliance. This hypothesis was supported by the observation that the ratio of clozapine/norclozapine, which has been used to identify CYP1A2 metabolizer phenotype or poor medication adherence depending on the value, was considerably higher in the current TDM population (2.2) vs the prior study on which the model was based (1.32). A higher ratio has been associated with either a poor CYP1A2 metabolizer phenotype or poor medication adherence. Additionally, in the TDM population the difference between the popPK model predicted and observed clozapine Cmin was strongly positively correlated with clozapine/norclozapine ratio but no other physiological covariate included in the popPK model. A post hoc sub‐
group analysis in individuals with a clozapine/norclozapine ratio < 1.5 further demonstrated a marked improvement in performance for the popPK model, emphasizing the relevance of poor medication adherence.
Overall, the data are clearly presented and modeled and the authors present a logical conclusion based on their observations, namely that TDM remains important to adequately control for environmental factors in Clozapine exposures, while modelling as presented here can help identify whether physiological vs environmental factors contribute to variability.
Major: However, I had some concerns about the differences in the two populations evaluated here and in the previous 2004 study on which the model was based. According to the Rostami-Hodjegan manuscript, "In 4963 (50%) samples (2222 patients), the plasma clozapine concentration was below the putative therapeutic threshold of 350 ng/mL." Whereas in the current TDM population, the authors here report that "One hundred and thirty‐nine patients (97.8 %) had a clozapine concentration within the accepted target concentration range (350 to 800 ng/mL)." (lines 183-184) I wonder if the authors could comment on this difference and its impact on their findings? One wonders whether in the current TDM population, whether the prescribing physicians have made more dose adjustments to keep patients in the therapeutic range and whether this is masking other physiological differences? In the original study, the population may not have had such fine monitoring and adjustments made, as evidenced by the large percentage of the population that is outside the therapeutic range. It is precisely those individuals that are outside the range for which the monitored physiological parameters may contribute the most to the observed clozapine concentrations. Is there any information available about dosing adjustments made in both studies?
Minor: In the introduction (lines 76-79), the authors state that the popPK model they are using from reference 21, was based on a cohort of 2,222 patients. In the actual reference by Rostami-Hodjegan, they report using 3782 patients in which 2222 showed clozapine concentrations below the therapeutic range. The authors should correct their statement or describe more fully if they were relying on a subset of the data in the 2004 paper.
Author Response
We thank the Reviewer for their time,
Please see the attached file
